# Knowledge of female genital cutting among health and social care professionals in Francophone Belgium: A cross-sectional survey

Sarah O'Neill[1]*, Fabienne Richard[1,2], Sophie Alexander[3], Isabelle Godin[1]

**1** CRISS, École de Santé Publique, Université libre de Bruxelles, Brussels, Belgium, **2** GAMS Belgique, Brussels, Belgium, **3** PERU, CR2, École de Santé Publique, Université libre de Bruxelles, Brussels Belgium

* sarah.oneill@ulb.be

## Abstract

The provision of optimal, equitable and gender-sensitive health-care to women and girls with FGM/C is challenging. Research indicates that health-professionals in receiving countries lack knowledge, confidence and competence in managing FGM/C. In order to develop policies that are suitable to the wide heterogeneity of women from FGM/C practising groups, it is paramount to identify what appropriate care might consist of and what are the knowledge gaps of health and social-care providers. The study-objective was to gain an understanding of the knowledge and practices of professionals working in the area of health in Francophone Belgium where no previous KAP has been done. An anonymous questionnaire with 24 questions was developed targeting healthcare and social-care professionals. The questionnaire was distributed between 6 May and 30 August 2021 via professional organisations for midwifery, social work, GPs, infectious disease specialists, paediatricians and gynaecologists. The results are presented as frequencies, differences in proportions between groups were tested with Pearson's Chi-square, when applicable. Correlations were tested by the Pearson correlation coefficient. The threshold of statistical significance is 5%. Four-hundred-and-sixty individuals filled in the questionnaire of which 42% were medical-doctors, 6% nurses, 27% midwives and 25% non-medical professionals (social-workers and psychologists). 55% of non-medical professionals had provided support for women with FGM/C. Almost 40% of health-professionals knew that there were 4 types of FGM and were able to correctly describe them, 15% were unable to correctly describe any of the 4 types. Those who had already provided care for women with FGM/C were more numerous to know that there were 4 types (52%). Two health-professionals had received requests to perform FGM/C. Twenty-seven midwives and medical-doctors had received requests for re-infibulation. Growing numbers of health and social-care professionals are providing care for women with FGM/C. However, knowledge of FGM/C is suboptimal. Continuous professional training is crucial.

**Data Availability Statement:** Data variable have been uploaded as supporting information.

**Funding:** The study was funded by GENDERNET Plus (to IG) in collaboration with partners at the

University of Montreal in Canada (Bilkis Vissandjée), at Paris 1 Pantheon-Sorbonne University in France (Armelle Andro) and the Department of women's and Children's Health, University of Upsala in Sweden (Birgitta Essen). https://www.era-learn.eu/network-information/networks/gender-net-plus/1st-joint-call- on-gender-and-un-sustainable-development-goals/sharing-actions-and-strategies-for- respectful-and-equitable-health-care-for-women-with-fgc-m. The funders had no role in study design, data collection and analysis, decision to publish, or preparation of the manuscript.

**Competing interests:** The authors have declared that no competing interests exist.

## Background

It is estimated that over 200 million women worldwide have been affected by some form of Female Genital Cutting, also called Female Genital Mutilation by the institutions of the United Nations. Some people strongly object to the term 'mutilation' for the modification of genitals for non-medical, cultural reasons. There are also activists who have undergone female genital cutting or worked with women who have who strongly believe in the importance of raising awareness about the harmfulness of the practice by calling the practice a 'mutilation'. From this point onwards we refer to the practice as FGM/C to include both of these groups. Different variations of FGM/C have been customarily practiced in at least 28 countries in Africa, parts of the Middle East (such as Iraq, Yemen), and Asia (such as Indonesia, Malaysia, and Dahwwodi Bohra of India); these countries are often referred to as "source countries". In most of these countries the practice is now illegal, but may continue. The persistence of the practice is linked to complex notions of social and gender identity and gender expectations, which imbues cut women with status and respectability within their families and inter-marrying groups. The extensive research that has been done over the last 40 years shows that in many places where FGM/C is a tradition, stopping the practice may bring serious social consequences with itself, varying from a bride being rejected upon marriage, to loss of status, dishonour, social exclusion, violence and other forms of stigmatisation [1]. FGM/C is internationally recognised as a form of gender-based violence based on deeply entrenched gender inequalities and there are sustained efforts to stop the practice[2,3]. Research shows that the socio-cultural and identity aspects of the practice can affect the psycho-social well-being of women, particularly when women migrate to places where the practice is not a social norm [1].

The World Health Organization classifies 'Female Genital Mutilation' into four major types. Type 1 is the partial or total removal of the clitoral glans and/or the prepuce. Type 2 is the partial or total removal of the clitoral glans and the labia minora, with or without removal of the labia majora. Type 3, also known as infibulation, is the narrowing of the vaginal opening through the creation of a covering seal. The seal is formed by cutting and repositioning the labia minora, or labia majora, sometimes through stitching, with or without removal of the clitoral prepuce and glans. Type 4 includes all other harmful procedures to the female genitalia for nonmedical purposes, for example, pricking, piercing, incising, scraping, and cauterizing the genital area.

FGM/C can involve immediate long-term psychological, sexual and physical health consequences [4]. Health problems are particularly severe for women who have undergone type III FGM/C–infibulation–as appropriate management of the condition requires de-infibulation before birth and sometimes before sexual intercourse [4]. In countries where infibulation is practiced, re-infibulation after childbirth is a common procedure. However, the closing of the woman's internal labia again after delivery to leave a small opening for urine and menstruations is also considered as female genital mutilation since it leads to the same health consequences. It is illegal in Belgium even if it is requested by women (article 409 of the penal code).

Through international migration thousands of women affected by FGM/C are now living in "receiving countries", mostly high-income countries. It has been estimated that in Europe over half a million women and girls live with FGM/C and that 180,000 girls and women are at risk of undergoing FGM/C every year [5].

FGM/C is outlawed in most EU countries; (for the countries where the KAP has been implemented: Belgium since 2001; France since 1979 and Sweden since 1982). Yet, legal commitments do not always translate into good health or sensitive health care practices and can, by stigmatizing the practice, unintentionally create obstacles to care [6].

Both in source and in receiving countries, provision of optimal, equitable and gender-sensitive health care to women and girls with FGM/C, and to those who may be at risk of being

exposed to this practice is challenging. Studies of health sector involvement in the management of FGM/C in receiving countries show variable availability of specialist services and staff training [6–8]. Research and systematic reviews also indicate that health professionals may lack knowledge, confidence and competence in managing FGM/C [6,9–11]. In order to develop policies that are suitable to the wide heterogeneity of women from FGM/C practising groups and their families, it is paramount to identify what the most appropriate care might consist of, and what are the health professionals', psychologists' and social workers' knowledge gaps.

Four countries (Belgium, Canada, France and Sweden) joined in a common research package with a general objective of improving care of migrants with FGM/C and cross-fertilizing the experiences. Each country developed a specific methodology to collect quantitative data on the knowledge, attitudes and practices of professionals confronted to women with FGM/C. In Belgium the objective was to explore both health and social-care providers. Although KAP studies among health professionals in Flanders had previously been undertaken [12,13], no previous research has specifically addressed the knowledge attitudes and practices of health professionals in Francophone Belgium and the bi-lingual Brussels region.

## Methods

### Ethics statement

The Erasme-ULB university clinic ethics committee waived the need for ethical approval for the study on the 8 March 2021 (Reference P2021/190). All research methods and data management were carried out in accordance with the guidelines of the European Commission for ethical research. Informed consent was obtained from all study participants. No personal identifiers were taken to ensure the confidentiality and privacy of study participants in compliance with EU ethics guidelines. All data are stored in a password protected data base by Isabelle Godin.

### Study setting

For funding reasons, the study took place in Francophone Belgium only (Wallonia-Brussels Federation), which constitutes approximately 40% of the Belgian population and includes Brussels, the capital city and Wallonia (South of Brussels). Anyone is allowed to consult French or Dutch-speaking services, but has to accept the official language of communication and administration of these services. As for professional organisations, members are free to join French or Dutch-speaking professional organisations depending on what language they prefer to communicate in.

Belgium is a country of immigration, so that, according to the most recent estimates, by December 2020, 93,685 girls and women from a country where female genital cutting/mutilation is practiced are residents [14]. Of these, 35,459 are affected or concerned by female genital mutilation (either already circumcised or at risk) including 12,730 minors (under 18 years). Based on the assumption that girls are intact if they arrived before age 5 and excised if arrived after 5 years, it is hypothesized that **23,395** excised girls and women live in Belgium and **12,064** are intact but at risk of excision if no preventive work is done. The five most represented countries of origin are Guinea, Somalia, Egypt, Ethiopia and Côte d'Ivoire. More than 16,500 girls and women who have undergone FGM/C or are at risk of undergoing it live in Flanders (the Dutch-speaking region), 10,000 in Brussels-Capital Region (bilingual region) and 8,800 in the Walloon Region (the French-speaking region). Every year 1,700 excised women give birth in a Belgian maternity hospital and require appropriate care [14].

Belgium has a national social security system for all residents with an added provision for undocumented persons. This means that all people have access to care [15].

## Study design

An anonymous questionnaire with 24 questions was developed by the authors in collaboration with colleagues at the Université de Paris 1 Panthéon-Sorbonne in France and the University of Montreal, Canada and from Upsala University (Sweden). The specific focus of this questionnaire was that it should target healthcare as well as social care professionals in Francophone Belgium. Although this study was designed to give an overview of FGM/C related knowledge, competences, attitudes and the need to undertake further training among professionals in Francophone Belgium, the collaboration with colleagues in France and Canada was meant both to broaden the perspective and to enable comparisons across countries. In Sweden, a study assessing the knowledge attitudes and practices (KAP) of health care providers on FGM/C had recently been undertaken by the Ministry of Health and therefore the Swedish team did not participate in the survey, while there was a need to implement such a survey in the other two countries. Specific objectives of this Belgian questionnaire were the following: (i) to assess the knowledge about FGM/C of health-, psychological- and social care providers (i.e. the WHO types, and prevalence); (ii) to assess whether the provider is able to identify a woman's medical, psychological and social needs (including de- infibulation, Post Traumatic Stress Disorder and referral to a specialist for specific needs or expertise); (iii) to describe how providers address and overcome communication barriers, information provision and referral to specialist clinics; (iv) to measure how providers identify risk of FGM/C, respond to it and identify what health and psycho-social care providers perceive to be indicators of risk. This information in turn will serve to identify priorities in terms of training or knowledge gaps of the professionals concerned. This study presents the results specifically on health and social care providers' knowledge on FGM/C. Data on attitudes and practices assessed in responses to case studies will be published elsewhere.

**Pretest.**   The pre-test of the questionnaire was piloted among health and social care professionals in Francophone Belgium between December 2020 and January 2021 (paper version of the questionnaire) and between March and April 2021 (electronic version on Lime Survey) to test for technical problems, defaults of clarity and to rectify inconsistencies.

## Survey distribution

The questionnaire was distributed via the following professional organisations: Union professionelle des Sages-femmes Belges (UPSfB) and Association Francophone des sages-femmes catholiques (AFSFC) for midwifery, Union des villes et des communes de Wallonie (UVCW) for social workers in the Belgian welfare centre, Federal Agency for Reception of Asylum Seekers (Fedasil) of the Brussels and Wallonie region for social workers working with asylum seekers; Office de la Naissance et de l'Enfance (ONE) the national agency for health and social care workers providing antenatal and postnatal care services; Société Scientifique de Médecine Générale (SSMG) for generalist practitioners; Centre Hospitalier Chrétien (CHC) for « infectious disease specialists and internal medicine »; Groupement Belge des Pédiatres de langue Française (GBPF) for paediatricians; Groupement des Gynécologues Obstétriciens de Langue Française de Belgique (GGOLFB) for gynaecologists. Each member of these organisations received by email with an invitation to link into a website where they could participate anonymously to the survey. The link to the electronic questionnaire was sent via email or the newsletter of these organisations with an informed consent information page explaining the larger study objectives, the institutions involved as well as ensuring the anonymity of the research participants. The questionnaire was distributed and filled in by consenting research participants between the 6[th] May and the 30[th] of August. The professional organisations did not to send out reminders.

**Table 1. Socio demographic characteristics of respondents.**

| Age | N | % |
|---|---|---|
| 20–39 years | 156 | 54,9 |
| 40–54 years | 67 | 23,6 |
| 55 years and + | 61 | 21,5 |
| Total | 284 | 100 |
| Sex | N | % |
| Men | 40 | 14,1 |
| Women | 243 | 85,9 |
| Total | 283 | 100 |
| Profession | N | % |
| Medical doctors | 166 | 42,2 |
| Nurses | 24 | 6,1 |
| Non-medical professionals | 97 | 24,7 |
| Midwives | 106 | 27 |
| Total | 393 | 100 |

**Data analysis.** The results are presented as frequencies, differences in proportions between groups were tested with Pearson's Chi-square, when applicable. Correlations were tested by the Pearson correlation coefficient. The threshold of statistical significance is 5%.

## Results

A total of 460 individuals filled in the questionnaire. However, there was almost no variable without missing values, which is why we present the numbers of respondents in each table. Significantly more women (86%) than men (14%) participated in the study. In terms of professions, 166 (42%) were medical doctors, 24 nurses (6,1%), 106 midwives (27%) and 97 non-medical professionals (24,7%) including social workers and psychologists (Table 1).

Almost 53% (199) of the research participants had already provided care or support for a woman or girl with FGM/C in contrast to 47% (178) who had not or were not sure if they had (Table 2). In terms of profession, 61% of medical doctors (n = 98) stated that they had not provided care for a woman or girl with FGM/C or were not sure if they had (Table 2). In terms of medical specialisation, 5 of these were gynaecologists, 64 general practitioners and 22 paediatricians. In contrast to this, 81% (n = 80) of midwives stated that they had already cared for women with FGM/C. Among other professionals, such as social workers and psychologists,

**Table 2. "Having provided care for a woman/girl with FGM/C" by age and profession (in %).**

| Age (N = 279, p = 0.05) | Yes | No/not sure |
|---|---|---|
| 20–39 yrs | 52,3 | 47,7 |
| 40–54 yrs | 66,7 | 33,3 |
| 55 and + | 46,6 | 53,4 |
| Total | 54,5 | 45,5 |
| Profession (N = 377, p<0,001) | Yes | No/not sure |
| Medical doctor | 39,1 | 60,9 |
| Nurse | 17,4 | 82,6 |
| Non medical | 55,3 | 44,7 |
| Midwife | 80,8 | 19,2 |
| Total | 52,8 | 47,2 |

Table 3. Care provided in the last 12 months by profession (in %, N = 196, n.s.).

| Profession | 0 | 1–5 times | 6–10 times | >10 times |
|---|---|---|---|---|
| Medical doctor | 23,3 | 58,3 | 8,3 | 10,0 |
| Nurse | 33,3 | 33,3 | 0,0 | 33,3 |
| Non medical | 38,5 | 48,1 | 5,8 | 7,7 |
| Midwife | 17,3 | 53,1 | 12,3 | 17,3 |
| Total | 25,0 | 53,1 | 9,2 | 12,8 |

psychotherapists, sex therapist, marriage counsellors, 55% (n = 52) stated that they had provided support for women with FGM/C in contrast to nurses, among which only 17% stated that they had provided support (Table 2). Seventy-seven percent of medical doctors and 83% of midwives stated having provided care for a patient with FGM/C over the last 12 months (Table 3).

The number of years of professional experience was not necessarily related to having consciously cared for a girl or woman with FGM/C. Forty-two percent (n = 31) of professionals with maximum 4 years of experience stated that they had cared for a woman or girl with FGM/C in contrast to 46% (n = 30) with more than 25 years of professional experience (pearson chi square < 0.01). Those with between 5 and 11 years of professional experience (63%, n = 42) and between 12 and 25 years of experience (68%, n = 51) were the most numerous to have cared for women or girls with FGM/C. Twenty-seven individuals have had a request for re-infibulation after childbirth, of which 9 medical doctors and 18 midwives. One medical doctor and one midwife stated that they were asked to perform a FGM/C on a girl.

## Competence and knowledge

Health and social care professionals were asked how many women living with FGM/C they estimated to reside in Belgium. Only 9% (n = 29) knew the correct figure, 39% said they did not know (n = 125) and 54% (n = 171) ticked a wrong answer. Health professionals were asked if they thought they were able to recognise FGM/C (Table 4). Thirty-three percent of the doctors felt that they would not be able to, 65% stated that they were uncertain whether they could and 5% were almost certain that they would be able to recognise FGM/C. In contrast to that 7% (n = 5) of midwives felt that they would not be able to recognise a FGM/C, 87% (n = 60) felt uncertain whether they were able to and 6% (n = 4) of midwives also felt that they were almost always able to recognise a FGM/C (differences between professions indicate a p value <0.01).

Health and social care professionals were asked if they knew into how many types of the WHO classified 'Female Genital Mutilation' and they were asked to describe the types (Table 5). Forty-eight respondents named four types and were able to correctly describe them; twenty-nine percent of those who knew that there were four types of FGM were only able to describe three correctly, 14% were able to correctly describe two, 3% were correctly able to

Table 4. Perceived ability to recognise and FGM by profession (in %, N = 191, p<0,01).

| Profession | Almost always | Uncertain | No |
|---|---|---|---|
| Medical doctor | 5,4 | 65,2 | 29,5 |
| Nurse | 0,0 | 80,0 | 20,0 |
| Midwife | 5,8 | 87,0 | 7,2 |
| Total | 5,2 | 73,8 | 20,9 |

**Table 5. Number of FGM types identified by WHO and number of FGM types correctly described (in%, N = 289).**

| WHO describes types of Female Genital Mutilation. How many do you know ? | | Number of types correctly described | | | | | |
|---|---|---|---|---|---|---|---|
| | | 0 | 1 | 2 | 3 | 4 | Total |
| | 1 | 42,3% | 57,7% | 0,0% | 0,0% | 0,0% | 100% |
| | | 11 | 15 | 0 | 9 | | 26 |
| | 2 | 12,2% | 53,7% | 34,1% | 0,0% | 0,0% | 100% |
| | | 5 | 22 | 14 | 0 | 0 | 41 |
| | 3 | 15,8% | 11,6% | 31,6% | 41,1% | 0,0% | 100% |
| | | 15 | 11 | 30 | 39 | 0 | 95 |
| | 4 | 14,9% | 2,5% | 14,0% | 28,9% | 39,7% | 100% |
| | | 18 | 3 | 17 | 35 | 48 | 121 |
| | 5 | 60,0% | 0,0% | 0,0% | 20,0% | 20,0% | 100% |
| | | 3 | 0 | 0 | 1 | 1 | 5 |
| | 6 | 0,0% | 100,0% | 0,0% | 0,0% | 0,0% | 100% |
| | | 0 | 1 | 0 | 0 | 0 | 1 |
| | Total | 18,0% | 18,0% | 21,1% | 26,0% | 17,0% | 100% |
| | | 52 | 52 | 61 | 75 | 49 | 289 |

describe one and 15% were unable to correctly describe any of the four WHO types. Twenty-six respondents thought there was only one type identified by WHO, among them, 58% could correctly describe one, and 42% of those who thought there was one type were not correctly able to describe what parts of the genitalia had been modified or removed.

Table 6 shows that younger age groups were more numerous in knowing that there were four WHO FGM types (48% among 20–39 year olds in contrast to 33% among 40 year olds and above, p<0.05). Those who had already provided care for women or girls with FGM/C were more likely to know that there were four types of FGM according to the WHO definition (52% of those who had provided care in contrast to 23% of those who had not or were not sure if they had, p<0.001).

Health and social care professionals were asked if they ever bring up the topic of FGM/C during their professional practice (Table 7). Forty-seven percent of medical doctors, 59% of nurses and 20% of midwives and 26% of social workers said that they never did. Of those health and social care professionals who did bring up the practice, the largest numbers did so if the woman/girl was originally from a FGM/C practising country (12% of medical doctors, 38% of midwives and 26% of non-medical professionals).

## Discussion

### Rate of exposure to FGM/C by profession

In the 2006 survey 58% of Flemish gynaecologists had already seen patients with FGM/C in their clinics [16]. Re-grouping all medical doctors regardless of specialisation, our francophone survey showed that 39% of doctors had provided care for a woman with FGM/C. In contrast

**Table 6. Cross-table: Ability to correctly describe the different types of FGM according to age (in %, N = 265, p<0,05).**

| Age groups | Wrong answer | Correct (4 types) |
|---|---|---|
| 20–39 yrs | 52,3 | 47,7 |
| 40 and + | 67,2 | 32,8 |
| Total | 58,9 | 41,1 |

**Table 7. Do you bring up FGM/C during your professional practice?.**

| | No, never | Yes but only if I suspect a risk of FGM for a girl | Yes, systematically except if the woman is accompanied by someone close | Yes, but only if the woman/family speaks about it first | Yes, systematically when a woman or girl is originally from a FGM practising country | Sometimes | Other |
|---|---|---|---|---|---|---|---|
| Medical doctor (n = 139) | 46,8% | 4,3% | 0,7% | 10,8% | 12,2% | 20,9% | 4,3% |
| Nurse (n = 17) | 58,8% | 5,9% | 5,9% | 5,9% | 0,0% | 17,6% | 5,9% |
| Non-medical (n = 78) | 25,6% | 1,3% | 2,6% | 21,8% | 25,6% | 19,2% | 3,8% |
| Midwife (n = 81) | 19,8% | 4,9% | 4,9% | 7,4% | 38,3% | 21,0% | 3,7% |
| Total (n = 315) | 35,2% | 3,8% | 2,5% | 12,4% | 21,6% | 20,3% | 4,1% |

to that 81% of midwives stated that they had already provided care for women with FGM/C. In Flanders the 2012–2013 survey showed that only 15,4% of midwives had seen a woman who had undergone FGM/C in the last 12 months. This represents a stark contrast to the figures for Brussels and Wallonia where 83% of midwives and 76% of medical doctors had seen between 1 and more than 10 women with FGM/C over the last 12 months (Table 3). It is uncertain why the figures differ so greatly, it is possible that some health and social care professionals who were frequently exposed to the practice responded to our survey in Francophone Belgium. In Flanders training on FGM/C for midwives was not provided before 2013 which may be why the figures were significantly lower in the 2012/2013 survey.

Although it has been repeatedly suggested that social workers have a crucial role to play in prevention work [17,18], little research on their views has been published to date. Fifty-five percent of non-medical social care professionals who participated in this survey stated that they had already provided care for women with FGM/C. It is possible that these figures are inflated because the questionnaires were sent out to social services in charge of vulnerable migrant populations (asylum seekers and refugees). As FGM/C and gender violence are recognised as valid reasons to grant asylum, it is important that social workers in these services are competent on the subject matter. An additional reason may be that social workers who may come into contact with women with FGM/C have been systematically trained since 2017 across the whole of Belgium for services who provide care and support for asylum seekers (communal public welfare centres (CPAS / OCMW), Fedasil and Rode Kruis). In French speaking Belgium these services have been provided with training on FGM/C for longer.

## Other determinants of exposure: Age

The figures on years of professional experience and having cared for women with FGM/C also seemed to differ between Wallonia and Flanders. Whereas in the Flanders survey, midwives below the age 30 had been more frequently confronted with FGM/C (18,6%) in contrast to older midwives (13,5%), in Francophone Belgium 67% of health and social care professionals between the ages of 40–54 stated that they had already provided care for women with FGM/C, in contrast to health and care professionals between the ages of 20–39 of which 52% had already cared for women who had undergone the practice. Even among the oldest age group, 55 years and above, 47% had already provided care for the population concerned. These figures indicate that FGM/C is an important issue that most health and social care professionals are

exposed to throughout their professional life in Francophone Belgium requiring good knowledge and training on how to provide optimal care.

## Knowledge and attitude

**Knowledge of prevalence.**   The majority of health and social care professionals stated that they did not know how many women with FGM/C resided in Belgium (39%) or ticked the wrong answer (53%). Whereas those medical professionals who may need to examine the intimate parts of a woman (midwives and medical doctors) tended to overestimate the number of women subjected to FGM/C in Belgium, non-medical professionals underestimated the figures. It is therefore crucial that they are trained on how to provide optimal care.

**Capacity to identify and categorize.**   Sixty-five percent of medical doctors stated that they were uncertain that they would be able to recognise a FGM/C, in contrast to 87% of midwives. Only 5% of medical doctors and 6% of midwives felt certain that they would be able to recognise a FGM/C. Yet, 40% of those who knew that there were four types of FGM according to the WHO definition were able to correctly describe them. Those who got the number of types wrong, thinking for instance that there were more or less than 4, were more likely to describe them wrongly. In contrast to this, a survey in the US showed that less than 28% of respondents could correctly identify all four types of FGM [19]. Our Francophone Belgian survey showed that those who had already provided care for women with FGM/C were also more numerous to know that there were four types (52%) in contrast to those who were not sure if they had provided care for a woman with FGM/C (23%), which shows that experience has an important role to play. In terms of age group, the 20–39 year olds were more numerous to know and correctly identify the four types (47,7% in contrast to 33% among the older age groups). There may be numerous reasons for this: because of migration, the number of women who have undergone FGM/C has quadrupled in Belgium between 2007 and 2020 and younger health professionals are more likely to have been exposed and to have received more information and training during their studies. It has also been observed in other studies that midwives have better FGM/C knowledge and diagnostic ability than medical doctors, (60.8% vs. 32.7%) [20].

**Requests for FGM/C and re-infibulation.**   Previous studies have addressed to what extent health professionals encountered requests to have FGM/C performed on girls and re-infibulation on mothers after childbirth. The 2019 US survey found that 3% of healthcare providers who participated in their survey had been asked to perform an FGM/C [19]. In contrast to this, the Swedish 2006 KAP showed that there were no requests for FGM/C to health professionals [21]. Our survey in Francophone Belgium showed lower figures on requests for FGM/C than the study by Leye et al in 2008 in Flanders. In Francophone Belgium two had received requests to perform FGM/C at some point throughout their professional career in contrast to Flanders 15 years ago where 31 gynaecologists had received such requests.

With regard to re-infibulation, the Francophone Belgian figures were slightly lower than in Flanders. Twenty-seven midwives and medical doctors had received requests for re-infibulation at some point during their professional practice in Francophone Belgium, compared with Leye's 2008 study in Flanders where 27% (34) of health professionals who had already cared for infibulated patients had received such requests. In Sweden in 2006, 19% (n = 25) of gynaecologists and 15% (n = 48) of midwives had received requests for re-infibulation. Differences in both requests for FGM/C and re-infibulation have decreased over the last 15 years between the two surveys, reflecting presumably, that things are changing, and migrants are informed that such procedures are illegal/ not performed in the EU.

**Discomfort with addressing the topic.**   Many health professionals declared that they never brought up the topic with the patient. This was even true for midwives (20%) who by

essence will be caring for the area where the FGM/C is performed. This issue of discomfort of professionals in addressing this topic has already been described, in France by Tantet et al, 2018 [11] and in Switzerland on HIV patients by Mauri et al, 2022 [22] and more generally by Evans et al 2019 [6] who discuss the consequences of lack of culturally sensitive care, including communication, in greater detail.

## Strengths and limitations

### Strengths

**Novelty of the study.** This was the first survey on health and social care professionals' knowledge, attitudes and practices regarding Female Genital Cutting in Wallonia-Brussels Federation, the Francophone part of Belgium. In Flanders, the Dutch-speaking part of Belgium two such surveys were previously undertaken on gynaecologists' in 2006 [16] and on midwives in 2012–2013 [13].

It was also, for receiving countries, the first survey to combine an assessment of the knowledge, attitudes and experiences of medical practitioners with different specialisations, midwives, nurses as well as social care professionals.

**Common research protocol.** Another strength is that this project is part of a larger research programme including Canada, France and Sweden, so that a similar methodology is in use.

## Limitations

### Sampling strategy meant that results are probably non representative of professions

Unfortunately, we had relatively few participants in this study, even if Belgium is a relatively small country (estimated population of 11,584,008 in 2022 of which approximately 4,5 million are Francophone). Some professions are better represented than others though the route chosen (web-based questionnaire sent via email by the professional organisation) was the same for all professionals. Recall bias is always a risk when asking health-professionals questions on the numbers of patients they have seen. It is also likely that those who already had some knowledge of the practice and were interested in the topic participated in the survey whereas those with no particular interest in the subject did not. These are the pitfalls that all surveys need to take into consideration.

**Cultural exposure to FGM/C of respondents.** We did not request the ethnic origin of the health and social care providers in the survey. Such questions are not common practice in Belgium and may be considered intrusive. We did not want to deter people from the survey by asking such questions and preferred to focus on individuals' professional knowledge and competence. It is possible that providers from FGM/C practising countries are more susceptible to the difficulties women with FGM/C may be experiencing. Had we had this information, comparison with US and UK data would have been possible. We therefore do not know if knowledge, competence, and "comfort" in discussing the topic on FGM/C may in any way be related to linguistic background and ethnic origin.

## Conclusion

This research shows that growing numbers of health and social care professionals are providing care for women with FGM/C. However, knowledge, attitudes and in particular comfort in addressing the topic with affected women is suboptimal. It is also a cause of concern that even care providers such as midwives for whom this is central to their professional practice are

reluctant to address the topic. Competence is not only theoretical knowledge such as the classification of types, but primarily enhancing verbal culturally sensitive communication to avoid "fear of addressing" or even "taboos" around the topic. This is crucial particularly for those who need to provide optimal care for the growing number of women living with FGM/C in Belgium.

## Supporting information

**S1 Text. Questionnaire.**
(PDF)

**S1 Data. Data variables.**
(XLSX)

## Acknowledgments

We owe enormous gratitude to our collaborators with whom hours were spent working on various drafts of the questionnaire online: in Canada, Bilkis Vissandjée, the PI of this Gendernet study; in France, Claire Tantet, Armelle Andro and Chemsa Le Coeur. We also thank Birgitta Essén in Sweden and Jasmine Abdulcadir in Switzerland for meetings, discussions and warm collaboration. We thank Morgan Vandegoor for the preparation of the questionnaire on Lime Survey.

We thank all health and social care professionals who participated in the study.

## Author Contributions

**Conceptualization:** Sarah O'Neill, Fabienne Richard, Sophie Alexander, Isabelle Godin.

**Formal analysis:** Sarah O'Neill, Isabelle Godin.

**Funding acquisition:** Fabienne Richard, Sophie Alexander, Isabelle Godin.

**Methodology:** Sarah O'Neill, Fabienne Richard, Sophie Alexander, Isabelle Godin.

**Supervision:** Isabelle Godin.

**Writing – original draft:** Sarah O'Neill.

**Writing – review & editing:** Fabienne Richard, Sophie Alexander, Isabelle Godin.

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
