## [Decision Letter · Decision Letter 0]

3 Oct 2023

PGPH-D-23-01271

Knowledge of Female Genital Cutting among health and social care professionals in Francophone Belgium: A cross-sectional survey

Dear Dr. O’Neill,

Thank you for submitting your manuscript to PLOS Global Public Health. After careful consideration, we feel that it has merit but does not fully meet PLOS Global Public Health’s publication criteria as it currently stands. Therefore, we invite you to submit a revised version of the manuscript that addresses the points raised during the review process.

The manuscript has been evaluated by two reviewers, and their comments are available below.

The reviewers have raised a number of major concerns. They feel the manuscript should outline a clearly-defined research question, and they request improvements to the reporting of methodological aspects of the study, for example, regarding the exclusion criteria and more information on how the data collection was completed. The reviewers also note concerns about the statistical analyses presented and request re-analyses be completed.

Could you please carefully revise the manuscript to address all comments raised?

We look forward to receiving your revised manuscript.

Kind regards,

Avanti Dey, PhD

Staff Editor

Journal Requirements:

1. Please provide additional details regarding participant consent. In the ethics statement in the Methods and online submission information, please ensure that you have specified (1) whether consent was informed and (2) what type you obtained (for instance, written or verbal, and if verbal, how it was documented and witnessed). If your study included minors, state whether you obtained consent from parents or guardians. If the need for consent was waived by the ethics committee, please include this information.

3. Please provide separate figure files in .tif or .eps format only and remove any figures embedded in your manuscript file. Please also ensure all files are under our size limit of 10MB.

4. We noticed that you used "unpublished" in the manuscript. We do not allow these references, as the PLOS data access policy requires that all data be either published with the manuscript or made available in a publicly accessible database. Please amend the supplementary material to include the referenced data or remove the references.

5. We have noticed that you have uploaded Supporting Information files, but you have not included a list of legends. Please add a full list of legends for your Supporting Information files after the references list.

Additional Editor Comments (if provided):

Reviewers' comments:

Reviewer's Responses to Questions

**Comments to the Author**

1. Does this manuscript meet PLOS Global Public Health’s publication criteria? Is the manuscript technically sound, and do the data support the conclusions? The manuscript must describe methodologically and ethically rigorous research with conclusions that are appropriately drawn based on the data presented.

Reviewer #1: Partly

Reviewer #2: Yes

2. Has the statistical analysis been performed appropriately and rigorously?

Reviewer #1: I don't know

Reviewer #2: Yes

3. Have the authors made all data underlying the findings in their manuscript fully available (please refer to the Data Availability Statement at the start of the manuscript PDF file)?

Reviewer #1: Yes

Reviewer #2: Yes

4. Is the manuscript presented in an intelligible fashion and written in standard English?

Reviewer #1: No

Reviewer #2: Yes

5. Review Comments to the Author

Reviewer #1: Thank you for the opportunity to review this paper. The issue of female genital cutting or female genital mutilation is an important area to explore and I commend the authors on this work. It is clear that with global migration more women who have experienced FGM are moving to other regions and so knowledge of FGM is important in all countries.

In the Abstract, I suggest writing a clear aim. This should also be in the end of the Background. I always like to see a statement that is “the aim of this study was to ….”

In terms of the sample, I couldn't work out why infectious diseases specialists and internal medicine specialists were surveyed. I was not sure of the relevance for those two disciplines but without a clear aim I cannot be sure.

The studies conducted in one part of Belgium, but this is a study in four countries Belgium, Canada, France and Sweden. I could not work out why it was only in one part of Belgium (Francophone Belgium) if it's a national program. Why wasn't in the other parts of the country, especially when a lot of the data presented in study setting is about the numbers of women in the whole of Belgium, who are at risk of have this procedure or have had this procedure.

Only five gynaecologists responded t the survey which feels like a significant problem.

Some of the phrases are not clear. In paragraph two on page 10 I don't understand what ‘having consciously accompanied a woman or a girl with FGM’ means. Does that mean caring for a woman. Later on that paragraph is another phrase – ‘ the most numerous to have accompanied women or girls with FGM’. I'm assuming that again, is about caring for so perhaps a bit of attention needs to be placed on the phraseology.

I was interested in the model of care that these women experienced in Francophone Belgium. Did they have continuity of care? Did they see midwives in the hospital in the community? Were they mostly seeing midwives or doctors?

In the beginning of the discussion, I was hoping to an overarching summary of the results before we delve into the different areas.

As a non European, I found use of Flemish/Flanders, Francophone Belgium terms and naming the different regions quite confusing. For those living in Europe, it's probably very straightforward, but I found it quite difficult to follow.

There are a number of limitations in the study but the authors do not detail these. The small number of participants in the study especially gynaecologists, is a significant issue. Recall bias is also a limitation. I think the survey probably attracted respondents who knew more about FGM so then the sample is quite biased.

I was surprised that there did not seem to be question about how to actually care for these women. The focus seemed to be on identification rather than actual care?

There are significant issues with the tables for tables. There are too many - nine tables and they look like they've just been taken the cross-tab output from SPSS or whatever statistical program and put into the tables. They're all formatted differently.

The references also were quite messy with uppercase titles and some of the journal abbreviations are not clear (eg Ref 12).

Some minor issues

In the abstract the acronym FGM/C is used, but it's not written out.

The same issue for KAP in the Background.

Reviewer #2: Knowledge of Female Genital Cutting among health and social care professionals in Francophone Belgium: A cross-sectional survey

The authors present findings of the research study titled “Knowledge of Female Genital Cutting among health and social care professionals in Francophone Belgium: A cross-sectional survey”.

This is an important paper that adds to the evidence on the knowledge level of health professionals and social care professionals in Belgium. The findings can be a strong basis for advocacy for structured training of the professionals to take lead in FGM prevention and response in Belgium and Europe where a lot of migrants with FGM related complications live.

6. PLOS authors have the option to publish the peer review history of their article (what does this mean?). If published, this will include your full peer review and any attached files.

**Do you want your identity to be public for this peer review?** For information about this choice, including consent withdrawal, please see our Privacy Policy.

Reviewer #1: No

Reviewer #2: No

---

## [Decision Letter · Decision Letter 1]

6 May 2024

PGPH-D-23-01271R1

Knowledge of Female Genital Cutting among health and social care professionals in Francophone Belgium: A cross-sectional survey

Dear Dr. O’Neill,

Thank you for submitting your manuscript to PLOS Global Public Health. After careful consideration, we feel that it has merit but does not fully meet PLOS Global Public Health’s publication criteria as it currently stands. Therefore, we invite you to submit a revised version of the manuscript that addresses the points raised during the review process.

The revised manuscript has been re-evaluated by two reviewers, and their comments are available below. Both reviewers are happy with the revisions to the manuscript and responses to their comments. They have suggested some minor revisions to improve the manuscript, including slight adaptions to the tables and textual revisions to clarify certain points. Please include any sample size or power calculations that were used when you resubmit your manuscript. If you did not calculate an appropriate sample size, please discuss the possible limitations of this.

We look forward to receiving your revised manuscript.

Kind regards,

Emma Campbell, Ph.D

Staff Editor

Journal Requirements:

Reviewers' comments:

Reviewer's Responses to Questions

**Comments to the Author**

1. If the authors have adequately addressed your comments raised in a previous round of review and you feel that this manuscript is now acceptable for publication, you may indicate that here to bypass the “Comments to the Author” section, enter your conflict of interest statement in the “Confidential to Editor” section, and submit your "Accept" recommendation.

Reviewer #1: All comments have been addressed

Reviewer #2: All comments have been addressed

2. Does this manuscript meet PLOS Global Public Health’s publication criteria? Is the manuscript technically sound, and do the data support the conclusions? The manuscript must describe methodologically and ethically rigorous research with conclusions that are appropriately drawn based on the data presented.

Reviewer #1: Yes

Reviewer #2: Yes

3. Has the statistical analysis been performed appropriately and rigorously?

Reviewer #1: Yes

Reviewer #2: Yes

4. Have the authors made all data underlying the findings in their manuscript fully available (please refer to the Data Availability Statement at the start of the manuscript PDF file)?

Reviewer #1: Yes

Reviewer #2: Yes

5. Is the manuscript presented in an intelligible fashion and written in standard English?

Reviewer #1: Yes

Reviewer #2: Yes

6. Review Comments to the Author

Reviewer #1: Thank you for addressing all my comments.

My only suggestion is the tables, They do look a lot better however I still think they could be combined and improved. I will leave it to the Editor to decide.

Reviewer #2: Knowledge of Female Genital Cutting among health and social care professionals in Francophone Belgium: A cross-sectional survey

The authors present findings of the research study titled “Knowledge of Female Genital Cutting among health and social care professionals in Francophone Belgium: A cross-sectional survey”.

This is an important paper that adds to the evidence on the knowledge level of health professionals and social care professionals in Belgium. The findings can be a strong basis for advocacy for structured training of the professionals to take lead in FGM prevention and response in Belgium and Europe where a lot of migrants with FGM related complications live.

The comments identified in the previous review have been addressed comprehensively.

The following TWO very minor comments need to be addressed as well

Abstract: Most of the comments were compressively addressed

1. Results section: “Those who had already provided care for FGM/C were more numerous to know that there were 4 types (52%). This statement is not clear, and I modify it to read “Those who had already provided care to women with FGM/C were more likely to know that there were 4 types (52%)”. Kindly adopt it to make it clear.

Methods

2. All the comments in the methods sections have been addressed comprehensively.

Results

3. The authors should remove the % symbol in the in texts with the tables but include it on the top row of each table to avoid repetition and redundancy.

7. PLOS authors have the option to publish the peer review history of their article (what does this mean?). If published, this will include your full peer review and any attached files.

**Do you want your identity to be public for this peer review?** For information about this choice, including consent withdrawal, please see our Privacy Policy.

Reviewer #1: **Yes: **Caroline Homer

Reviewer #2: No

---

## [Editor Report · Decision Letter 2]

11 Jun 2024

Knowledge of Female Genital Cutting among health and social care professionals in Francophone Belgium: A cross-sectional survey

PGPH-D-23-01271R2

Dear Dr O’Neill,

We are pleased to inform you that your manuscript 'Knowledge of Female Genital Cutting among health and social care professionals in Francophone Belgium: A cross-sectional survey' has been provisionally accepted for publication in PLOS Global Public Health.

Best regards,

Julia Robinson

Executive Editor